# Do Community Home-Based Elderly Care Services Improve Life Satisfaction of Chinese Older Adults? An Empirical Analysis Based on the 2018 CLHLS Dataset

**DOI:** 10.3390/ijerph192315462

**Published:** 2022-11-22

**Authors:** Zhao Zhang, Yihua Mao, Yueyao Shui, Ruyu Deng, Yuchen Hu

**Affiliations:** 1College of Civil Engineering & Architecture, Zhejiang University, Hangzhou 310027, China; 2Center for Balance Architecture, Zhejiang University, Hangzhou 310028, China; 3Binhai Industrial Technology Research Institute, Zhejiang University, Tianjin 300301, China

**Keywords:** community home-based elderly care, life satisfaction, Chinese older adults

## Abstract

Population aging has become a major challenge for the Chinese government. Based on the Chinese Longitudinal Healthy Longevity Survey (CLHLS) in 2018, this study adopts the propensity score matching (PSM) method to assess the effect of community home-based elderly care services (CHECS) on the life satisfaction of the elderly in China. The results demonstrate that CHECS can improve their life satisfaction. Compared with life care services (LCS) and medical care services (MCS), the positive effect of spiritual and cultural services (SCS) and reconciliation and legal services (RLS) is more obvious. Moreover, the heterogeneity test demonstrates that the effect is more significant for the elderly who live with their families, whose activities of daily living are unrestricted, and whose depression levels are lower. The results obtained indicate that CHECS need precise policies for different elderly groups, attention to the positive impact of SCS and RLS on the life satisfaction of the elderly, and the substantive effectiveness of LCS and MCS.

## 1. Introduction

Life satisfaction, an overall assessment of an individual’s quality of life based on predefined criteria [1], is a subjective cognitive process of comparing the current situation with pre-defined criteria [2]. A specific heterogeneity in the criteria of life satisfaction exists for people in different life cycles [3]. Under the trend of accelerated population aging, on the one hand, the transformation of social roles and the decline of physical functions of the elderly enable them to begin to reassess their own values. Psychological problems such as low self-esteem, loneliness, loss, depression, and paranoia gradually emerge and bring about a reduction in mobility and the deterioration of the physical health of the elderly [4,5]. On the other hand, with the change in social structure and the weakening of family functions, it is increasingly difficult to satisfy the physical and mental needs of the elderly by depending solely on care provided by children and relatives. The intergenerational support from children to parents in a family with fewer children may also increase the psychological burden of the elderly on hampering their children’s work [6,7]. A deterioration of physiological functions and an increase in psychological burdens negatively affect the daily lives of the elderly and reduce their life satisfaction. In this context, numerous researchers have explored the influencing factors for the life satisfaction of the elderly and investigated effective ways to improve their quality of life from different perspectives [8,9].

The seventh census indicates that 13.5% of China’s population is over 65 years old, and the country has fully entered an aging society [10]. Therefore, it is urgent to promote a senior care service system in line with China’s national conditions. From the perspective of the existing elderly care model in China, the family elderly care model is still the main one, and the institutional elderly care model is supplementary. However, these two models are gradually showing shortages and weaknesses, and the community home-based elderly care model has been put forward. The differences and connections between three kinds of elderly care models are summarized in Table 1.

From the perspective of the institutional model, unlike the elderly in developed countries and regions, the Chinese elderly generally have a strong affection for family retirement. Moreover, because economic development and social welfare are still at a preliminary stage, the limited development and management capacity of institutional elderly care has led to various problems of “bullying the elderly” [11]. This has caused the public to question the institutional elderly care model. From the perspective of the family elderly care model, although this type of model can meet the home sentiment of Chinese older people, the “4-2-1” or “4-2-2” family form has led to the need for children to balance family elderly care, child-rearing, and work. Therefore, their children are under too much pressure from elderly care, and the weakness of the family elderly care model has gradually emerged [12].

Combined with the experience of mature elderly care models in developed countries, community home-based elderly care can meet the psychology of the elderly aging at home, relieve the pressure on their children, and allow the elderly to receive medical and elderly care services provided by relevant service institutions and professionals. This can combine the advantages of family and institutional elderly care to the greatest extent and become a novel idea for the development of the elderly care model integrated with Chinese characteristics [13]. Therefore, it is important to investigate the effectiveness of community home-based elderly care services (CHECS) and analyze the impact of CHECS on their life satisfaction to promote their healthy aging and improve the “home-based, community-depended, institutional-supplemented, medical combined elderly care service system” [14].

Previous studies have investigated the effects of CHECS on the quality of life of the elderly from different aspects [15,16]. At least three aspects need improvement. The first is in terms of research methods. Most prior studies employed traditional linear regression methods such as Logit, Probit, and Tobit and investigated whether they received CHECS as a dummy variable in the regression equation. This allowed for the comparison of the differences in quality of life between older adults who received CHECS and those who did not [17,18]. However, such methods overlook the heterogeneity of the two types of older adults and cannot overcome the biased estimation and sample selectivity deviation caused by sample “self-selection”, thus resulting in invalid results. The second aspect is in terms of research content. The majority of studies examined the impact of CHECS on the quality of life of older adults from the perspective of both physical and mental health [19,20]. However, only a few studies explored the impact on life satisfaction. The third aspect is in terms of the selection of control variables. Most studies only selected the individual characteristics and living habits of the elderly as control variables [21,22]. However, in China, which is still dominated by family retirement, financial, health, and emotional support from family has a significant impact on the elderly [23]. Only a few studies include these as relevant control variables, such as Chen and Hao’s study on the mental health of the elderly [24] and Yao et al.‘s study on the physical and mental health of the elderly [25]. In addition, based on Maslow’s needs theory, human life needs five levels, namely, physiological, security, love/belonging, respect, and self-actualization [26]. Among them, physiological and safety are primary needs, love/belonging and respect are intermediate needs, and self-actualization is a high need. When the lower-level needs are satisfied, the higher-level needs will subsequently be created [27]. Compared with physiological and psychological health, life satisfaction is a higher-level need. Therefore, the level of physiological and psychological health of the elderly affects their life satisfaction. However, only a few relevant studies include them as control variables.

In order to investigate the effect of different kinds of CHECS on the life satisfaction of Chinese older adults, this work employs propensity score matching (PSM), a quasi-natural experimental research method, to establish a counterfactual research framework based on data from the Chinese Longitudinal Healthy Longevity Survey (CLHLS) in 2018. We also empirically analyzed the effect of CHECS on their life satisfaction in different groups, with different activities of daily living (ADL), depression levels, and living conditions. The findings obtained are useful for broadening the research scope of CHECS on the quality of life of Chinese older adults. We aim to provide an important practical reference to further optimize CHECS supply, promote the development of a senior care service system in line with China’s national conditions, and contribute to active and healthy aging.

## 2. Materials and Methods

### 2.1. Participants

The data used in this study were derived from the cross-sectional data of the Chinese Longitudinal Healthy Longevity Survey (CLHLS) by the PKU Center for Healthy Aging and Development in 2018. The survey primarily covers the basic condition, evaluation of the current situation, personality and emotional characteristics, lifestyle, ADL, personal background, family structure, and physical health level of the elderly. In addition, the survey covers 85% of the regional scope of the country, involving a total of 500 sample areas in 22 provinces, municipalities, and autonomous regions, including Beijing. All study participants had the same conditions for the analysis. CLHLS data have the advantages of strong authority, wide coverage, and a relatively high number of survey participants. According to the definition of the elderly population and the age distribution of the data, the primary research object is the elderly population aged 60 years and above. Sample data of those under 60 years old were excluded, and the missing values and unanswerable values in all sample data were also excluded. The final number of valid samples obtained was 3796.

### 2.2. Variable Description

#### 2.2.1. Dependent Variable

This study measures the life satisfaction of the elderly by the “evaluation of the current situation.” According to the question “How do you think your life is now?” in the CLHLS questionnaire, the variable of life satisfaction of the elderly was assigned as 1 for very bad, 2 for bad, 3 for average, 4 for good, and 5 for very good.

#### 2.2.2. Independent Variables

CHECS refer to the provision of various services and assistance to the elderly in the community. This includes daily care, medical care, and spiritual comfort. According to the question “What social services are available for the elderly in your community?”, there are eight categories: “daily care, visiting the doctor (medicine delivery), spiritual comfort (chatting and relieving boredom), daily shopping, organizing social and recreational activities, providing legal aid (protection of rights), providing health care knowledge, dealing with family and neighborhood disputes,” and the options are “yes” and “no”. Taking into account the differences in service contents, CHECS were subdivided into four categories [28]: medical care services (MCS), including home visits to the doctor (delivery of medicine), and the provision of health care knowledge, both of which were not available and were assigned a value of 0. Those with one or two of these services represented a community with MCS and were assigned a value of 1. The following three categories of services were similarly handled: life care services (LCS), including personal care and daily shopping; spiritual and cultural services (SCS), including spiritual comfort (chatting to relieve boredom) and organization of social and recreational activities, including social entertainment; and reconciliation and legal services (RLS), including the provision of legal aid (rights protection) and handling family and neighborhood disputes.

#### 2.2.3. Control Variables

Drawing on existing studies on the factors influencing the life satisfaction of Chinese older adults, 21 variables in four areas—personal characteristics [22], lifestyle habits [22], physical and mental health level [29], and family support status of older adults [23]—were selected as control variables.

In terms of personal characteristics, two types of control variables—natural attributes and socioeconomic status—were introduced in this study. Age and years of education were continuous variables. Gender, household registration, whether living with a spouse, whether living with family, and whether having commercial insurance or social security were dichotomous variables, with values of 1 for male and 0 for female, 1 for urban and 0 for rural, 1 for living with their spouse and 0 for not, 1 for living with family and 0 for not, and 1 for having commercial insurance or social security and 0 for not. The relative economic level was divided into “very rich, relatively rich, average, relatively poor, and poor”, with values from 5 to 1.

With regard to lifestyle habits, prior studies have demonstrated that people can improve their physical and mental health by forming and maintaining healthy habits [30]. The more positive attitudes older people have toward life in old age, the more stable their health behaviors are, and the higher their quality of life will be. We chose the questions “Do you smoke regularly?”, “Do you drink alcohol regularly?”, “Do you exercise regularly?”, and “Do you have annual medical examinations?”, with values of 1 for yes and 0 for no, as well as the variable “sleep quality”, which was assigned 5 for very good, 4 for good, 3 for average, 2 for bad, and 1 for very bad to indicate lifestyle habits.

In terms of physical and mental health levels, the multidimensional character of health determines the diversity of health measurement indicators. This work measures the health level of the elderly from two dimensions: physical health and mental health, based on the study by Tao et al. [31]. First, for physical health, this work selects three indicators for multidimensional measurement, namely, self-rated health, activities of daily living (ADL), and illness from the perspectives of subjective evaluation and objective assessment. Self-rated health is a subjective indicator that can comprehensively reflect the individual health status and plays a positive predictive role in the risk of morbidity and mortality of the elderly [32]. In this work, the self-rated health variables were defined as 5 for very good, 4 for good, 3 for fair, 2 for bad, and 1 for very bad. ADL, the most basic measure of the health of the elderly, including “dressing, bathing, eating, getting in and out of bed, going to the toilet, and bowel control”, is defined as the ability to perform activities of daily living during the last six months. In addition, it is also an effective approach to evaluating the health level of the elderly by examining their disease status. Twenty-four common diseases of the elderly, including heart disease and diabetes, were considered in the CLHLS, and the disease status of the elderly was defined as 1 if they have a certain disease, and 0 if they did not. The total score was derived by summing up the 24 diseases. Secondly, the mental health of the elderly was evaluated based on two aspects: the depression level and personality and emotion. The CES-D scale has been widely employed to measure mental health, as it has been well-documented to have high validity, internal consistency, and acceptable retest stability [33]. CES-D has been widely applied to measure mental depression and has good validity in studies on Chinese samples [34]. The scale has the same four options for all six questions on depression in CLHLS, five of which are negative statements and one positive. In this study, the five negative statement questions were transformed into positive statements, and the response options of six were summed to obtain the CES-D score. Respondents’ depression scores were taken as integer values of [6,30], with lower scores indicating more severe depression. Similarly, the respondent’s personality–emotional score was given a value of [7,35], with lower scores indicating more severe negative emotions.

Finally, in terms of family support, the three primary dimensions of financial support, health support, and emotional support from the family were considered. Among them, financial support mainly refers to the material exchange between families. Based on the question “How much cash did your children (including all grandchildren and their spouses who live with you and not living with you) give you in the past year?”, the variable “family financial support” was constructed and assigned the value of 1 for provided and 0 for not. The variable “family emotional support” was constructed based on the question “Who do you tell first if you have something in your mind?”. If the elderly person confides in their spouse and children (including all grandchildren and their spouses who live together), the family was considered to provide emotional support and was assigned a value of 1. Otherwise, it was assigned a value of 0. The variable “family health support” was constructed based on the question “Who takes care of you when you are not feeling well or when you are sick?”. If the elderly were taken care of by their spouses and children (including all grandchildren and their spouses living together or not living together), then the family was considered to provide health support and was assigned a value of 1. Otherwise, it was assigned a value of 0.

### 2.3. Methodology

Compared with traditional linear regression methods, the PSM method can effectively overcome the “selection bias” caused by biased estimation and sample “self-selection” [35]. Since PSM does not require prior assumptions about the functional form, parameter constraints, and error term distribution, nor does it require the explanatory variables to be strictly exogenous, it has advantages in addressing the endogeneity of the treatment variables. Therefore, this work adopts this method for model estimation and empirical analysis, which is performed in the following four steps.

In the first step, covariates were selected. Drawing on the relevant literature, the factors affecting the life satisfaction of Chinese older adults and the supply of CHECS were included in the model, namely, personal characteristics, lifestyle habits, physical and mental health levels, and family support status, to ensure that the negligibility assumption was met.

In the second step, the propensity scores were calculated. In this study, we applied the Logit model to compute the propensity score value for the individual to receive CHECS.

In the third step, PSM was performed. (1) The matching method was selected. It is well known that there is no superiority or inferiority in matching methods, but various matching methods have particular measurement biases. Therefore, even when processing the same sample data, different measurement results may be generated. No consensus was reported by the academic community on which matching method should be employed to optimize the results. However, if the results after applying multiple matching methods were similar or consistent, the matching results were robust and the sample validity was good [36]. Therefore, to enhance the reliability of the research findings, k-nearest neighbor matching, radius matching, and kernel matching were used for matching. (2) The balance was tested. If the propensity scores were estimated more accurately, a standardized deviation could be employed to assess whether the matched distribution between the treatment and control groups achieved statistical data balance.

In the fourth step, the average treatment effect was computed. The average treatment impact comprises three categories. The first is the average treatment effect (ATT) of the treatment group, which is the average change in the life satisfaction of the elderly who received the community elderly home care service. The second is the average treatment effect (ATU) of the control group, which is the average change in the life satisfaction of the elderly who did not receive the community elderly home care service. The third is the average treatment effect (ATE) of the total sample, which is the mean of the change in the life satisfaction of the random sample of the elderly. Since this study explores the contribution of community home care services to the life satisfaction of the elderly, focusing on those who received community home care services, ATT is more appropriate for the analysis.

## 3. Results

### 3.1. Descriptive Statistics

The minimum, maximum, mean, and standard deviation statistics of each variable were computed (Table 2). The mean value of life satisfaction, self-rated health, and depression level among the survey respondents was 3.946, 3.495, and 22.718, respectively. This indicates that the overall life satisfaction, physical health level, and mental health level of the elderly were all high. Further analysis of the CHECS provided for the elderly revealed that the coverage of elderly services was narrow, and some of the services were low in content and accessibility, which could not effectively meet their needs. The statistics from the questionnaire indicate that nearly half of the communities where the elderly resided provide MCS. Other CHECS were rarely provided, with only 14.1% of senior communities providing LCS, 26.2% providing SCS, and 34.6% providing RLS. In terms of lifestyle habits, Chinese older adults have fewer smoking and drinking habits, and sleep well, while nearly 70% of Chinese older adults have annual medical examinations but rarely participate in positive aging behaviors that require high physical mobility, such as exercising. Finally, it is worth noting that most children still live with the elderly and offer them health, financial, and emotional support under the traditional family concept of “filial piety” culture. Family aging is still an important way of aging for the elderly in China.

### 3.2. Overlap Test

The common support hypothesis requires that the propensity scores of the treatment and control groups have a common range of values. To ensure the matching quality of the sample data, the kernel density function plots were further plotted after deriving the propensity scores to assess the common support domain after matching, as shown in Figure 1, Figure 2, Figure 3 and Figure 4. The propensity scores of the sample receiving the CHECS and the sample not receiving these services have an extensive range of overlap. Most of the observed values were within the common range of values. Therefore, it can be assumed that the matching effect is ideal, and the common support hypothesis was satisfied.

### 3.3. Balance Test

To ensure the reliability of the propensity score matching results, this work draws on Lian et al.’s work [37] and adopts the mean examination balance hypothesis. Table 3 lists the mean *t*-tests of the matched variables for the four CHECS types. Based on the *t*-values, after matching, no significant systematic difference in the covariates was reported between the control and treatment group, except for the difference in life satisfaction.

### 3.4. Average Effect Analysis

This study measured the average treatment effect of four types of CHECS provision on the life satisfaction of the elderly. The estimation results after matching with three different methods (Table 4) were consistent, indicating that the sample data have good robustness. Therefore, the arithmetic mean of the effects was chosen to characterize the effects for the subsequent empirical analysis.

After the counterfactual estimation of PSM, the impact of LCS on Chinese older adults’ life satisfaction was insignificant for all three matching methods. MCS significantly affected Chinese older adults’ life satisfaction only in the kernel match, with a net effect of 0.046. This indicates that access to MCS contributes to a significant increase in Chinese older adults’ life satisfaction of 0.046, after accounting for Chinese older adults’ selectivity bias. SCS and RLS significantly affect the life satisfaction of Chinese older adults in all three matches. The ATT for the treatment group of SCS was 0.060, indicating that access to SCS significantly increased life satisfaction by 0.060 when other factors were excluded. The ATT for the treatment group of RLS was 0.080, indicating that access to RLS significantly increased life satisfaction by 0.080 when other factors were excluded. The model results indicated that the three types of CHECS, namely, MCS, SCS, and RLS, could significantly improve the life satisfaction of the elderly, in the order of: RLS (ATT = 0.080) > SCS (ATT = 0.060) > MCS (0.046). LCS had no significant effect on the life satisfaction of Chinese older adults.

### 3.5. Heterogeneous Effect Analysis

Due to the different levels of physical health, mental health, and living conditions, the needs of various types of CHECS vary considerably [38]. In the prior study, the ATT of the treatment group was chosen to measure the net effect of CHECS on the life satisfaction of the elderly. However, the ATT can only reflect the mean value of the change in life satisfaction of the elderly who received CHECS but cannot reflect the structural differences in the effect of the elderly sample. Thus, exploring the heterogeneous effect of various types of older adults can enrich the existing literature on the welfare effects of CHECS on Chinese older adults. In this work, the sample was grouped and processed by using the ADL, the depression level, and whether the elderly lived with their families as markers to assess the group differences of the effect of four types of CHECS on their life satisfaction. The comparison results are shown in Table 5.

Heterogeneity tests demonstrate that, for Chinese older adults with restricted ADL, higher levels of depression, and those living on their own, the effects of all four types of CHECS on their life satisfaction were not significant under all three matching methods. For the elderly with unrestricted ADL, all three types of services, except for LCS, significantly increased their life satisfaction under the three matching methods, in the order of RLS (ATT = 0.116) > SCS (ATT = 0.088) > MCS (ATT = 0.064). SCS and RLS significantly improved the life satisfaction of Chinese older adults with low depression levels, with the degree of impact being SCS (ATT = 0.082) > RLS (ATT = 0.062). For Chinese older adults living with their families, all three types of services, except for MCS, significantly increased their life satisfaction, with the degree of impact being RLS (ATT = 0.084) > SCS (ATT = 0.075) > MCS (ATT = 0.071).

## 4. Discussion

Based on the analysis of the average effect, LCS could not improve the life satisfaction of Chinese older adults. This may be because the companionship and care by community workers can improve the physical and mental health of the elderly [39] but cannot replace the care and concern from their families [40]. Compared with older adults in developed countries, Chinese older adults care more about the “feeling of home” and look forward to sharing the joy of family with their offspring. The sense of belonging that family brings to Chinese older adults is stronger, which also reflects the vital need for affection in later life [41]. Additionally, due to the high workforce and professional requirements inherent in LCS, the current supply level and service quality of LCS are low and cannot meet the needs of the elderly in their daily lives.

The significant and low impact of MCS on Chinese older adults’ life satisfaction only in the kernel match may be related to the lower quality of MCS. Although a proportion of Chinese older adults with milder chronic diseases choose and receive community home-based MCS, when Chinese older adults have significant illnesses or are more limited in their daily living activities the majority still prefer to receive more specialized medical treatment in hospitals or elderly services, while only a few of them will choose community-provided MCS [42].

SCS and RLS significantly increased the life satisfaction of Chinese older adults in all three matching approaches, with a high degree of impact. This may be due to the fact that, on the one hand, with the rapid socioeconomic development, the family space has been extended and children are away from their parents for various reasons [43]. Meanwhile, retirement has removed the elderly from their original work environments, and changing social roles have brought an unprecedented sense of isolation [5]. Additionally, due to the difference in cognitive level and social adaptability, Chinese older adults are more likely to suffer from legal incidents such as fraud [44]. Therefore, their demands can easily be satisfied with SCS and RLS. On the other hand, both services require less energy and professionalism from community workers. Based on the existing community management foundation of “neighborhood committees” in China, these two services are more easily developed.

Based on the analysis of the heterogeneous effect, the four types of CHECS did not have a significant impact on the life satisfaction of the elderly with restricted ADL. MCS, SCS, and RLS had a significant impact on the life satisfaction of the elderly with unrestricted ADL. Based on Maslow’s theory of needs, only when the lower-level needs are satisfied can the higher-level needs subsequently arise [27]. For the elderly with restricted ADL, they need LCS the most, and other services do not help much to improve their life satisfaction. The current quantity and quality of LCS provided by the community are a little bit lower and cannot meet the needs of the elderly in their daily lives. This also indicates that LCS in China needs to be improved to better achieve the expected goal of promoting CHECS. Regarding depression levels, the four types of CHECS had no significant effect on the life satisfaction of Chinese older adults with higher depression levels. SCS and RLS had a significant impact on the life satisfaction of Chinese older adults with lower levels of depression. The degree of impact of SCS was higher than that of RLS. This may be because Chinese older adults with higher levels of depression are prone to the idea that “the elderly are useless” and tend to fall into self-denial [45], close themselves off, and limit their participation in community activities. Their families should offer more care and attention to the elderly and actively seek help from professional counselors to ensure that they become happier as they age. Conversely, Chinese older adults with unrestricted ADL and lower levels of depression were able to participate more actively in community-provided recreational activities, health talks, and other activities since they were more capable of taking care of themselves, and these activities enriched their spiritual life and enhance their life satisfaction.

In terms of living conditions, the four types of CHECS had no significant effect on the life satisfaction of the elderly living alone. LCS, SCS, and RLS had a significant impact on the life satisfaction of Chinese older adults living with their families. The elderly living alone, lacking a warm family atmosphere, felt helpless and lonely, and their emotions could not be satisfied. Although CHECS improve the living conditions of the elderly in all aspects, they cannot replace the companionship and comfort of family members [40] and thus do not have a significant effect on the life satisfaction of Chinese older adults living alone. In contrast, elderly people who do not live alone have the company of their spouses or children and spend their old age in a familiar environment, which is in line with their expectations. Additionally, frequent contact with family members has a protective effect on their psychological health [46]. After satisfying the need for children’s companionship, the services provided by the community further enrich the lives of the elderly and enhance their life satisfaction.

## 5. Conclusions

This study explored the impact of four types of CHECS on the life satisfaction of Chinese older adults, namely, LCS, MCS, SCS, and RLS. The results indicate that MCS, SCS, and RLS had varying degrees of improvement in their life satisfaction. However, the effect of LCS on their life satisfaction was insignificant. Next, using cohort difference analysis, this work then explored the heterogeneous impact of the four types of CHECS on the life satisfaction of the elderly in three categories: whether the ADL were limited, the level of depression, and whether they lived with their families. The four types of CHECS had a more prominent effect on the life satisfaction of Chinese older adults who lived with their children, whose daily living activities were not limited, and whose depression level was generally lower. The findings offer an essential reference for the Chinese government that CHECS need precise policies for different elderly groups, attention to the positive impact of SCS and RLS on the life satisfaction of the elderly, and the substantive effectiveness of LCS and MCS.

However, several limitations in our study could be improved in future research. First of all, due to the limitations of cross-sectional data, we could not discuss the long-term effect of CHECS on the life satisfaction of the elderly. Secondly, we attenuated the endogeneity problems caused by sample self-selection by PSM and adding control variables. However, endogeneity brought by missing variables is still unavoidable in our study. Therefore, a further collection of panel data for two-stage least squares (2SLS) or difference in difference (DID) analysis is a direction worth paying attention to in follow-up research.

## Figures and Tables

**Figure 1 ijerph-19-15462-f001:**
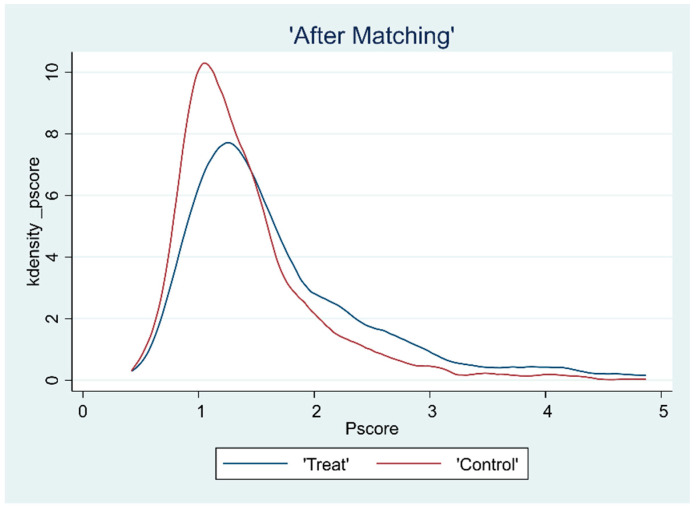
Kernel density function plot (LCS).

**Figure 2 ijerph-19-15462-f002:**
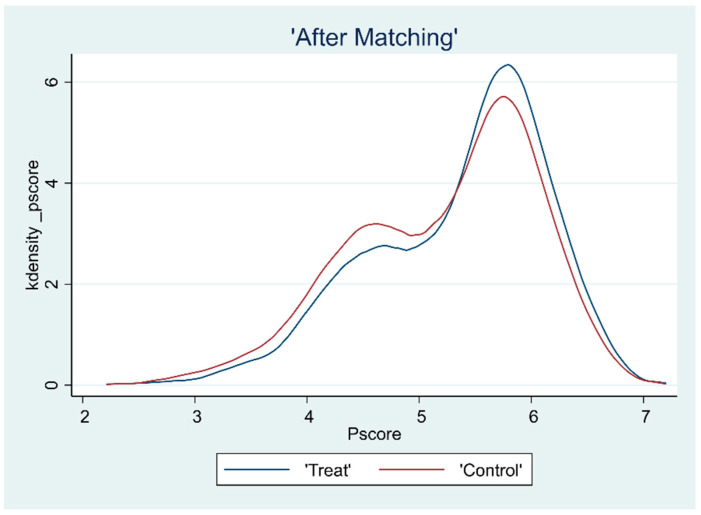
Kernel density function plot (MCS).

**Figure 3 ijerph-19-15462-f003:**
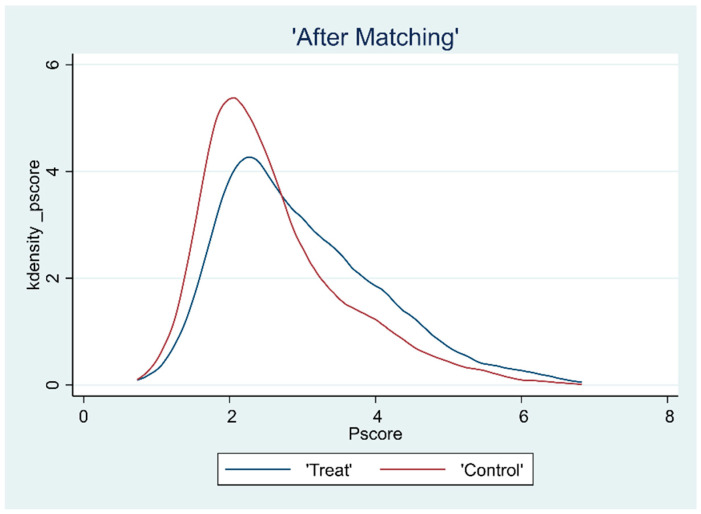
Kernel density function plot (SCS).

**Figure 4 ijerph-19-15462-f004:**
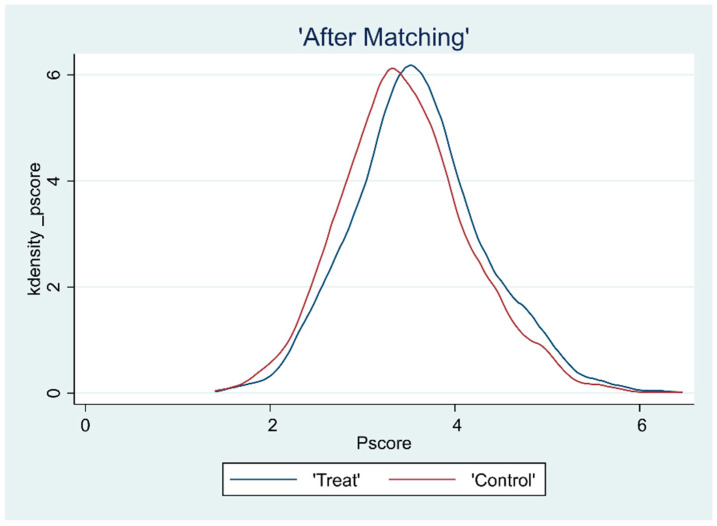
Kernel density function plot (RLS).

**Table 1 ijerph-19-15462-t001:** Differences and connections between three kinds of elderly care models.

	Where to Live	Services Provider	Cost
Family elderly care model	Home	Family members	Low
Institutional elderly care model	Institutions	Professional nursing staff	High
Community home-based elderly care model	Home	Family members and professional nursing staff	Relatively low

**Table 2 ijerph-19-15462-t002:** Descriptive statistics of variables.

Variable Category	Variable	Min.	Max.	Mean	S.D.
Dependent variable	Life satisfaction	1	5	3.946	0.804
Independent variables	Life care services (LCS)	0	1	0.141	0.348
	Medical care services (MCS)	0	1	0.523	0.500
	Spiritual and cultural services (SCS)	0	1	0.262	0.440
	Reconciliation and legal services (RLS)	0	1	0.346	0.476
Personal characteristics	Gender	0	1	0.471	0.499
	Age	60	117	83.212	11.299
	Household registration	0	1	0.179	0.383
	Years of education	0	18	3.796	4.364
	Relative economic level	1	5	3.135	0.652
	Whether to live with spouse	0	1	0.463	0.499
	Whether to live with family	0	1	0.835	0.371
	Availability of commercial insurance and social security	0	1	0.074	0.261
Living habits	Sleep quality	1	5	3.552	1.004
	Smoking or not	0	1	0.166	0.372
	Drinking or not	0	1	0.162	0.369
	Whether to exercise regularly	0	1	0.382	0.486
	Whether annual medical examination	0	1	0.693	0.462
Level of physical and mental health	Self-assessed health level	1	5	3.495	0.913
	ADL	1	3	2.611	0.632
	Illness	0	24	1.490	1.598
	Depression level	6	30	22.718	3.929
	Personality–emotion	13	35	27.078	3.435
Family support	Family financial support	0	1	0.621	0.485
	Family health support	0	1	0.933	0.250
	Family emotional support	0	1	0.916	0.277

**Table 3 ijerph-19-15462-t003:** Results of balance test.

Variables	Sample Matching	LCS		MCS		SCS		RLS	
		Deviation	*t*-Value	Deviation	*t*-Value	Deviation	*t*-Value	Deviation	*t*-Value
Gender	Unmatched	−0.005	−0.20	0.014	0.84	0.010	0.56	0.003	0.15
	Matched	0.001	0.04	0.006	0.38	−0.001	−0.07	0.004	0.22
Age	Unmatched	0.014	0.03	−0.161	−0.44	−0.596	−1.43	−0.644	−1.67 *
	Matched	−0.095	−0.14	0.031	0.09	−0.004	−0.01	−0.043	−0.10
Household registration	Unmatched	0.103	5.78 ***	0.010	0.77	0.124	8.83 ***	0.056	4.26 ***
	Matched	0.020	0.74	0.002	0.18	0.015	0.75	0.013	0.83
Years of education	Unmatched	1.045	5.15 ***	0.203	1.43	1.447	9.09 ***	0.668	4.50 ***
	Matched	0.239	0.81	0.039	0.28	0.133	0.62	0.138	0.78
Relative economic level	Unmatched	0.172	5.68 ***	0.080	3.79 ***	0.152	6.37 ***	0.093	4.19 ***
	Matched	0.027	0.64	0.027	1.31	0.020	0.67	0.017	0.66
Whether to live with spouse	Unmatched	−0.010	−0.43	0.021	1.28	0.034	1.85 *	0.038	2.21 **
	Matched	0.001	0.03	−0.000	−0.01	−0.001	−0.04	0.004	0.21
Whether to live with family	Unmatched	0.016	0.91	0.027	2.21 **	0.032	2.31	0.012	0.91
	Matched	0.008	0.35	0.001	0.11	0.003	0.17	−0.001	−0.09
Availability of commercial insurance and social security	Unmatched	0.010	0.81	−0.027	−3.15 ***	−0.001	−0.07	−0.016	−1.83 *
	Matched	0.002	0.15	−0.001	−0.12	−0.001	−0.04	0.000	0.03
Sleep quality	Unmatched	0.036	0.77	0.031	0.96	0.090	2.43 **	0.031	0.91
	Matched	0.003	0.04	0.013	0.40	−0.006	−0.14	0.006	0.16
Smoking or not	Unmatched	−0.010	−0.58	−0.029	−2.43 **	−0.040	−2.88 ***	−0.031	−2.47 **
	Matched	−0.004	−0.16	−0.002	−0.13	−0.004	−0.23	0.001	0.06
Drinking or not	Unmatched	0.003	0.17	0.010	0.84	0.010	0.76	0.006	0.46
	Matched	0.002	0.07	0.005	0.47	0.002	0.10	0.002	0.15
Whether to exercise regularly	Unmatched	0.095	4.18 ***	0.012	0.78	0.109	6.12	0.052	3.12 ***
	Matched	0.020	0.64	0.001	0.04	−0.002	−0.09	0.005	0.26
Whether receiving an annual medical examination	Unmatched	0.049	2.27 **	0.114	7.66 ***	0.023	1.38	0.053	3.35 ***
	Matched	0.007	0.26	0.001	0.05	−0.002	−0.11	0.003	0.16
Self-assessed health level	Unmatched	0.113	2.66 **	0.064	2.15 **	0.084	2.50 **	0.078	2.50 **
	Matched	0.009	0.16	0.020	0.70	−0.015	−0.38	0.006	0.17
ADL	Unmatched	−0.002	−0.08	−0.010	−0.47	−0.014	−0.58	0.022	1.02
	Matched	−0.003	−0.07	−0.012	−0.61	−0.004	−0.14	−0.002	−0.07
Illness	Unmatched	0.192	2.57 **	0.082	1.58	0.372	6.35 ***	0.211	3.88 ***
	Matched	0.047	0.46	0.023	0.45	0.019	0.24	0.045	0.70
Depression level	Unmatched	0.426	2.32 **	0.083	0.65	0.141	0.97	0.055	0.41
	Matched	0.074	0.31	−0.020	−0.16	−0.021	−0.12	0.000	0.00
Personality–emotion	Unmatched	0.490	3.06 ***	0.120	1.08	0.533	4.22 ***	0.225	1.93 *
	Matched	0.072	0.35	0.027	0.24	−0.014	−0.09	0.002	0.01
Family financial support	Unmatched	−0.044	−1.96	0.026	1.65 *	−0.031	−1.73 **	0.029	1.77 *
	Matched	−0.006	−0.21	0.007	0.42	−0.002	−0.07	0.000	0.00
Family health Support	Unmatched	−0.077	−6.61 ***	0.008	1.01	−0.025	−2.71 **	0.008	0.95
	Matched	−0.013	−0.66	−0.004	−0.51	−0.005	−0.37	−0.001	−0.11
Family emotional support	Unmatched	−0.037	−2.90	0.027	2.96 ***	0.002	0.19	0.022	2.36 **
	Matched	−0.003	−0.13	−0.002	−0.27	0.000	0.02	0.001	0.07

Note: ***, **, and * indicate that the estimation results are significant at the 1%, 5%, and 10% levels.

**Table 4 ijerph-19-15462-t004:** Results of average effect analysis.

Service Category	Matching Method	Treatment Group	Control Group	ATT	S.D.	*t*-Value
LCS	K-nearest neighbor matching	4.062	4.055	0.007	0.043	0.15
	Radius matching	4.058	4.030	0.028	0.039	0.74
	Kernel matching	4.062	4.017	0.045	0.038	1.17
	Mean	4.061	4.034	0.027	0.040	
MCS	K-nearest neighbor matching	3.992	3.945	0.047	0.029	1.63
	Radius matching	3.992	3.953	0.039	0.027	1.43
	Kernel matching	3.992	3.946	0.046	0.027	1.72 *
	Mean	3.992	3.948	0.044	0.028	
SCS	K-nearest neighbor matching	4.076	4.016	0.060	0.034	1.79 *
	Radius matching	4.076	4.019	0.057	0.031	1.88 **
	Kernel matching	4.076	4.012	0.064	0.030	2.13 **
	Mean	4.076	4.016	0.060	0.032	
RLS	K-nearest neighbor matching	4.039	3.948	0.091	0.030	2.99 ***
	Radius matching	4.039	3.966	0.068	0.028	2.66 **
	Kernel matching	4.039	3.962	0.077	0.028	2.79 ***
	Mean	4.039	3.959	0.080	0.029	

Note: ***, **, and * indicate that the estimation results are significant at the 1%, 5%, and 10% levels.

**Table 5 ijerph-19-15462-t005:** Results of heterogeneous effect analysis.

Service Category	Matching Method	ADL		Depression Level		Whether to Live with Families	
		Restricted	Unrestricted	CES-D > 20	CES-D ≤ 20	Yes	No
LCS	K-nearest neighbor matching	0.027 (0.34)	0.049 (0.99)	0.004 (0.08)	0.030 (0.34)	−0.059 (−0.51)	0.043 (0.95)
	Radius matching	0.010 (0.14)	0.042 (0.91)	0.013 (0.30)	0.080 (1.02)	−0.074 (−0.70)	0.051 (1.23)
	Kernel matching	0.013 (0.17)	0.053 (1.16)	0.026 (0.62)	0.083 (1.06)	−0.058 (−0.54)	0.071 * (1.74)
	Mean	0.016	0.048	0.014	0.065	−0.064	0.055
MCS	K-nearest neighbor matching	−0.042 (−0.76)	0.064 * (1.86)	0.041 (1.28)	0.040 (0.69)	0.047 (0.58)	0.033 (1.05)
	Radius matching	−0.028 (−0.54)	0.062 ** (1.98)	0.037 (1.26)	0.057 (1.06)	0.025 (0.35)	0.041 (1.42)
	Kernel matching	−0.015 (−0.30)	0.067 ** (2.12)	0.039 (1.31)	0.060 (1.12)	0.033 (0.45)	0.045 (1.54)
	Mean	−0.028	0.064	0.039	0.053	0.035	0.040
SCS	K-nearest neighbor matching	0.013 (0.21)	0.080 ** (2.05)	0.083 ** (2.32)	−0.028 (−0.40)	−0.06 (−0.66)	0.065 * (1.82)
	Radius matching	−0.003 (−0.05)	0.090 ** (2.55)	0.080 ** (2.46)	0.004 (0.05)	−0.06 (−0.69)	0.079 ** (2.41)
	Kernel matching	−0.000 (−0.00)	0.095 ** (2.72)	0.084 ** (2.60)	0.003 (0.05)	−0.06 (−0.72)	0.083 ** (2.55)
	Mean	0.004	0.088	0.082	−0.007	−0.06	0.075
RLS	K-nearest neighbor matching	−0.050 (−0.87)	0.1153 *** (3.24)	0.052 (1.56)	0.050 (0.82)	−0.023 (−0.28)	0.078 ** (2.41)
	Radius matching	−0.023 (−0.43)	0.114 *** (3.50)	0.064 ** (2.11)	0.066 (1.19)	0.037 (0.50)	0.085 *** (2.86)
	Kernel matching	−0.014 (−0.26)	0.120 *** (3.70)	0.070 ** (2.31)	0.073 (1.34)	4.072 (−0.03)	0.089 *** (3.03)
	Mean	−0.029	0.116	0.062	0.063	0.004	0.084

Note: ***, **, and * indicate that the estimates are significant at the 1%, 5%, and 10% levels, with significant *t*-values in parentheses.

## Data Availability

Chinese Longitudinal Healthy Longevity Survey (CLHLS) belongs to public database (https://opendata.pku.edu.cn/dataverse/CHADS (accessed on 17 October 2022)). Users can download relevant data for free for research and publish relevant articles. Our study is based on open source data, so there are no ethical issues or other conflicts of interest.

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
