# Peer review of "Do Community Home-Based Elderly Care Services Improve Life Satisfaction of Chinese Older Adults? An Empirical Analysis Based on the 2018 CLHLS Dataset"

_ijerph, 2022, doi:10.3390/ijerph192315462_

Round 1

Reviewer 1 Report

The manuscript needs a minor review. 

Author Response

Point 1: Authors should add some potential future work in the discussion or conclusion part.

Response 1: Thank you very much for your comments. We add the limitations and potential future work of our research in the conclusion part. First of all, due to the limitations of cross-sectional data, we cannot discuss the long-term effect of community home-based elderly care services (CHECS) on the life satisfaction of the elderly. Secondly, we attenuated the endogeneity problems caused by sample self-selection by propensity score matching (PSM) and adding control variables. However, endogeneity brought by missing variables is still unavoidable in our study. Therefore, a further collection of panel data for two stage least square (2SLS) or difference in difference (DID) analysis is the direction worth paying attention to in the follow-up research.

Reviewer 2 Report

Dear authors, I thank you for the opportunity to review your manuscript entitled: "Do Community Home-based Elderly Care Services Improve 2 Life Satisfaction of Chinese Older Adults? - An Empirical Analysis Based on the 2018 CLHLS Dataset".

Additional updates are required before its publication:

Introduction part:

A reference needs to be added - line 71, page 2.

Given that you mention previous studies, it is necessary to add references lines 74-76, page 2

In line 87 you write, "Only a few studies include these as relevant control variables." Since this is a specific statement, you need to indicate which studies these are, with references.

At the end of the introduction, I would suggest that you state the objectives of your study to make it clearer to the reader

Materials and Methods part:

It is not clear enough from your description of the sample whether all study participants had the same conditions for the analysis, since the study was conducted in 22 provinces

I suggest replacing the word Methodology (line 213, page 5) with the word "Analysis"

Discussion part:

I suggest that you delete the paragraph you mentioned at the beginning, that is, put it in the introductory part lines 342-346, page 12. 

In this section, you should list the main results of your study and their implications for practice.

Does this study have any limitations? If any, I suggest that you indicate them

Author Response

Point 1: A reference needs to be added - line 71, page 2.

Response 1: Thank you very much for your comments. According to your comments, we add the reference in line 71, page 2.

Point 2: Given that you mention previous studies, it is necessary to add references lines 74-76, page 2

Response 2: According to your comments, we add the reference in line 75, page 2.

Point 3: In line 87 you write, "Only a few studies include these as relevant control variables." Since this is a specific statement, you need to indicate which studies these are, with references.

Response 3: We add the references and statements of related studies, such as Chen and Hao's study on the mental health of the elderly and Yao et al. 's study on the physical and mental health of the elderly.

Point 4: At the end of the introduction, I would suggest that you state the objectives of your study to make it clearer to the reader

Response 4: We state the objectives of your study at the beginning of the last paragraph in the introduction part. “In order to investigate the effect of different kinds of CHECS on the life satisfaction of Chinese older adults, this work employs propensity score matching (PSM), a quasi-natural experimental research method, to establish a counterfactual research framework based on data from the Chinese Longitudinal Healthy Longevity Survey (CLHLS) in 2018.”

Point 5: It is not clear enough from your description of the sample whether all study participants had the same conditions for the analysis since the study was conducted in 22 provinces.

Response 5: We are so sorry for the missing statement of “the same conditions for the analysis”, and we have already made it up.

Point 6: I suggest replacing the word Methodology (line 213, page 5) with the word "Analysis"

Response 6: Thank you very much for your comments. Different from the specific analysis of data in the results part, this part is mainly an introduction to the propensity score matching (PSM) method. Therefore, we would like to remain the word Methodology (line 213, page 5).

Point 7: I suggest that you delete the paragraph you mentioned at the beginning, that is, put it in the introductory part lines 342-346, page 12. 

Response 7: Thank you very much for your comments. According to your comments, we delete the paragraph you mentioned at the beginning.

Point 8: In this section, you should list the main results of your study and their implications for practice.

Response 8: Thank you very much for your comments. We list the main results of our study in the conclusion part and supplement their implications for practice. “This study explored the impact of four types of CHECS on the life satisfaction of Chinese older adults, namely, LCS, MCS, SCS, and RLS. The results indicate that MCS, SCS, and RLS had varying degrees of improvement in their life satisfaction. However, the effect of LCS on their life satisfaction was insignificant. Next, using cohort difference analysis, this work then explored the heterogeneous impact of the four types of CHECS on the life satisfaction of the elderly in three categories: whether the ADL were limited, the level of depression, and whether they lived with their families. The four types of CHECS had a more prominent effect on the life satisfaction of Chinese older adults who lived with their children, whose daily living activities are not limited, and whose depression level is generally lower. The findings offer an essential reference for Chinese government that CHECS need: precise policies for different elderly groups; attention to the positive impact of SCS and RLS on the life satisfaction of the elderly; and the substantive effectiveness of LCS and MCS.”

Point 9: Does this study have any limitations? If any, I suggest that you indicate them.

Response 9: Thank you very much for your comments. We add the limitations and potential future work of our research in the conclusion part. First of all, due to the limitations of cross-sectional data, we cannot discuss the long-term effect of community home-based elderly care services (CHECS) on the life satisfaction of the elderly. Secondly, we attenuated the endogeneity problems caused by sample self-selection by propensity score matching (PSM) and adding control variables. However, endogeneity brought by missing variables is still unavoidable in our study. Therefore, a further collection of panel data for two stage least square (2SLS) or difference in difference (DID) analysis is the direction worth paying attention to in the follow-up research.

Reviewer 3 Report

-Generally, I would urge the authors to do another proofread of the article to ensure that any typos/mispellings are addressed in English.

-In the second paragraph, the authors talk about the "elderly care model," "the family elderly care model," and "the institutional elderly care model" without defining them. I think the authors should define these terms for the readers, especially as they are central to our understanding of community home-based elderly care services (CHECS), at the cornerstone of the paper.

-Overall, the Materials and Methods, Results, Discussion, and Conclusions are well written and developed, and I think this is a great paper to publish on the topic of Community Home-based Elderly Care Services in China.

Author Response

Point 1: In the second paragraph, the authors talk about the "elderly care model," "the family elderly care model," and "the institutional elderly care model" without defining them. I think the authors should define these terms for the readers, especially as they are central to our understanding of community home-based elderly care services (CHECS), at the cornerstone of the paper.

Response 1: Thank you very much for your comments. In order to help the readers to understand the differences and connections between 3 kinds of elderly care models, we add a table to define "the family elderly care model", "the institutional elderly care model", and  "the community home-based elderly care model" in the introduction part.

Table 1. Differences and connections between 3 kinds of elderly care models.

Where to live

Services provider

Cost

Family elderly

care model

Home

Family members

Low

Institutional elderly

care model

Institutions

Professional nursing staff

High

Community home-based elderly care model

Home

Family members and professional nursing staff

Relatively low

Round 2

Reviewer 1 Report

No comment.

Reviewer 2 Report

Many thanks for your effort in addressing the comments and submitting the revised manuscript. 

Reviewer 3 Report

Other than reviewing the English writing and editing it during the proofs process, I do not have additional comments for the authors.